# Myc beyond Cancer: Regulation of Mammalian Tissue Regeneration

**Barbara Illi *** and **Sergio Nasi ***

Institute of Molecular Biology and Pathology, National Research Council, c/o Department of Biology and Biotechnology "Charles Darwin", Sapienza University of Rome, 00185 Rome, Italy

* Correspondence: barbara.illi@cnr.it (B.I.); 44nasi@gmail.com (S.N.); Tel.: +39-064-991-2227 (B.I.)

**Abstract:** Myc is one of the most well-known oncogenes driving tumorigenesis in a wide variety of tissues. From the brain to blood, its deregulation derails physiological pathways that grant the correct functioning of the cell. Its action is carried out at the gene expression level, where Myc governs basically every aspect of transcription. Indeed, in addition to its role as a canonical, chromatin-bound transcription factor, Myc rules RNA polymerase II (RNAPII) transcriptional pause–release, elongation and termination and mRNA capping. For this reason, it is evident that minimal perturbations of Myc function mirror malignant cell behavior and, consistently, a large body of literature mainly focuses on Myc malfunctioning. In healthy cells, Myc controls molecular mechanisms involved in pivotal functions, such as cell cycle (and proliferation thereof), apoptosis, metabolism and cell size, angiogenesis, differentiation and stem cell self-renewal. In this latter regard, Myc has been found to also regulate tissue regeneration, a hot topic in the research fields of aging and regenerative medicine. Indeed, Myc appears to have a role in wound healing, in peripheral nerves and in liver, pancreas and even heart recovery. Herein, we discuss the state of the art of Myc's role in tissue regeneration, giving an overview of its potent action beyond cancer.

**Keywords:** tissue regeneration; gene expression; stem cells; somatic reprogramming

## 1. Introduction

Regeneration is a complex, energy-demanding process that is essential for organisms' survival, as it normally occurs in homeostasis or during the replacement of damaged or lost tissues. This process strictly resembles embryonic development, but it is not identical, as some molecular features diverge [1]. In humans, different tissues have diverse regeneration capacity, which depends mainly on the loss and damage entity [2,3]. Liver, blood, gut, skin, the endothelium and skeletal muscle are examples of tissues that regenerate throughout life. Conversely, the nervous (both central and peripheral) system and the heart possess limited reparative capacity. This characteristic of terminally differentiated tissues provides the background for the onset of diseases typical in aged individuals, such as heart failure and neurodegeneration (when this latter lacks genetic bases).

Regeneration requires the execution of complex gene expression programs able to orchestrate reconstitution from cell precursors, which differ in their origin according to the tissue to which they belong. Indeed, skin, blood and skeletal muscle are provided with stem cells niches which undergo cell division and differentiation upon homeostatic or injury signals. Other tissues, such as liver and the endothelium, rely on tissue-specific cell types for self-duplication and repair.

Myc is a basic helix–loop–helix–lecine zipper (bHLHZip) transcription factor (TF) [4,5] that controls pivotal cell functions, such as proliferation, apoptosis [6–9], metabolism [10–12] and senescence [13,14]. Of note, it is highly important for embryonic stem cells' (ES) self-renewal [15] and somatic cell reprogramming. Indeed, it is one of the four Yamanaka factors [16,17], which are able to revert a differentiated phenotype into an ES-like ground

state. Myc exerts its function at the gene expression level, but it is no longer considered as a canonical transcription factor which is bound to chromatin at promoters/enhancers, recruiting transcriptional co-activators and, thus, allowing RNA Polymerases (RNAPs)-dependent transcription to start. Surprisingly, Myc has been found to regulate all the three RNAPs; it has been demonstrated to control RNAPII-dependent activities, such as pause–release [18], elongation [19] and termination [20], as well as to ensure splicing fidelity [21], to influence chromatin topology [22] and to indirectly regulate transcription controlling the expression of microRNAs (miRNAs) [23]. For these reasons, any perturbation of Myc function has a profound impact on cell behavior. This is particularly evident in cancer cells, where Myc is frequently deregulated and overexpressed [24]. In healthy cells, Myc is tightly controlled and essential for survival; indeed, Myc-targeted anti-cancer therapies have to face the detrimental effects that decreased Myc activity may produce in normal cells. Tissue-specific resident stem cells need Myc to maintain their self-renewal and to proliferate; furthermore, reactivation of Myc-dependent transcriptional programs has been found as of major importance for the regeneration of tissues and for strategies aimed at counteracting the tissue degeneration typical of aging. In this review paper, we discuss the role Myc has in stem cell biology and, consequently, in tissue regeneration and repair, further underpinning its importance as a master regulator of cell functions beyond cancer.

## 2. Overview of Myc-Dependent Transcription

*Myc* is represented by three paralogs: *c-myc*, *n-myc* [25] and *l-myc* [26]. The c-*myc* (hereafter simply myc) gene is embedded in chromosome 8 long arm (8q24) and encodes a 439 amino-acid protein with a molecular mass of 64KD. Myc structure contains two highly conserved Myc polypeptides at the N-terminal and C-terminal regions. The N-terminal transactivating domain (TAD) holds three conserved sub-domains called Myc boxes (MB0, MBI and MBII) [27,28]. MBI recruits the cyclin CDK-complex positive transcription elongation factor b (P-TEFb), which phosphorylates RNAPII, allowing transcriptional elongation [18]. MBII rules the activation and repression of Myc target genes, promotes cellular transformation and is involved in Myc turnover. MBIII binds histone deacetylase 3 (HDAC3), promoting transcriptional repression, whilst MBIV is supposed to govern Myc pro-apoptotic activity. The C-terminal domain is provided with a 100-amino-acid bHLH-LZ domain, which represents the docking site for another small bHLH-LZ protein called Max, [29]. The Myc/Max interaction is stable. The heterodimer directly binds DNA "Enhancer boxes" (E-boxes) to induce transcription [30]. A proline-, glutamic acid-, serine- and threonine-rich region (PEST) is located across MBIII and MBIV boxes and two nuclear localization sequences are present within the MBIV and C-terminal basic region (Figure 1).

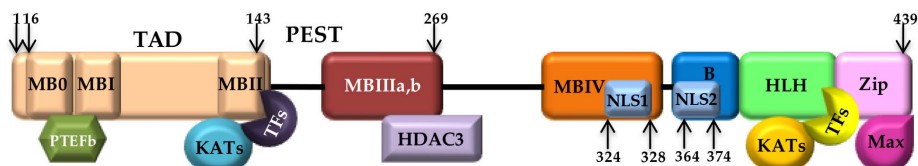

**Figure 1.** Domain structure of Myc protein. Schematic representation of Myc and its major interacting partners. Abbreviations: MB = Myc box; KAT = lysine (K) acetyltransferase; TF = transcription factor; HDAC3 = histone deacetylase 3; P-TEFb = positive transcription elongation factor b.

Interestingly, Myc has an intrinsically disordered structure. Intrinsically disordered-structure proteins (iDPs) are a challenge for scientists for a number of reasons [31]. Indeed, they cannot be studied with the standard methods used for folded proteins and present consistent deviation from the paradigm that structure = function. However, it is widely accepted that iDPs have biological roles that depend on their structure/dynamics and on their interaction with folded domains. Myc is basically organized in loose modules with unpacked secondary structure [32]. This is the main reason why the pharmacological inhibition of Myc is barely attainable, as it lacks pocket domains which may allocate small

molecules. Specifically, the Myc bHLH-LZ domain is highly disordered and a conventional globular structure is only assumed when bound to Max (see below) [33] and DNA [34]. When looking at cancer cells, the only chance to interfere with Myc function is to perturb its binding with protein partners, especially with Max [35]. Indeed, a number of papers reported the screening and the selection of a number of Myc-/Max-interfering small molecules [36–38], whereas the Myc mutant termed Omomyc [39] has been proven to be highly effective in inhibiting Myc/Max heterodimerization and Myc binding to cognate DNA elements [40]. Most recently, other approaches have been attempted, although less efficient, such as the use of bromodomain and extraterminal domain inhibitors (BETi) to switch off Myc expression [41] and promote inhibition of Myc-interacting kinases (e.g., Aurora Kinase A) [42].

### 2.1. The Myc (Max) Network

Given the requirement of Max for Myc to exert its transcriptional activation properties, Max has to be considered as a hub for the so-called Myc network. Max is a ubiquitous, stable TF. It has a short half-life, which makes its activity strictly dependent on the amount of associated TFs. Max forms homodimers, competes with Myc/Max heterodimers for the binding to E-boxes and associates with the bHLH-LZ domain, which contains families of TFs Mad1-4 (Mad1, Mxi1, Mad3 and Mad4), Mnt and Mga. These latter are very similar to Myc, as they poorly homodimerize and bind DNA, whereas they form strong heterodimers with Max. The Mad1 and Mxi1 families of transcriptional repressors compete with Myc/Max for DNA binding. These TFs are able to interact with co-repressors, such as Sin3a and Sin3b, and recruit HDACs and other chromatin remodelers to repress transcription. A shift in the balance between Myc/Max and Mad/Max heterodimers is dictated by the amount of Mads within the cell, determining cell fate decision and a repositioning from proliferation/transformation to differentiation/quiescence [43]. From this evidence, it appears that Myc and Mad act in opposite ways to activate/repress transcription by competing for Max and for the binding to E-boxes [44] (Figure 2). In spite of the repressor function of Mad/Max heterodimers, Myc itself can specifically repress transcription by replacing p300 and binding the transcriptional activator Miz-1 [45].

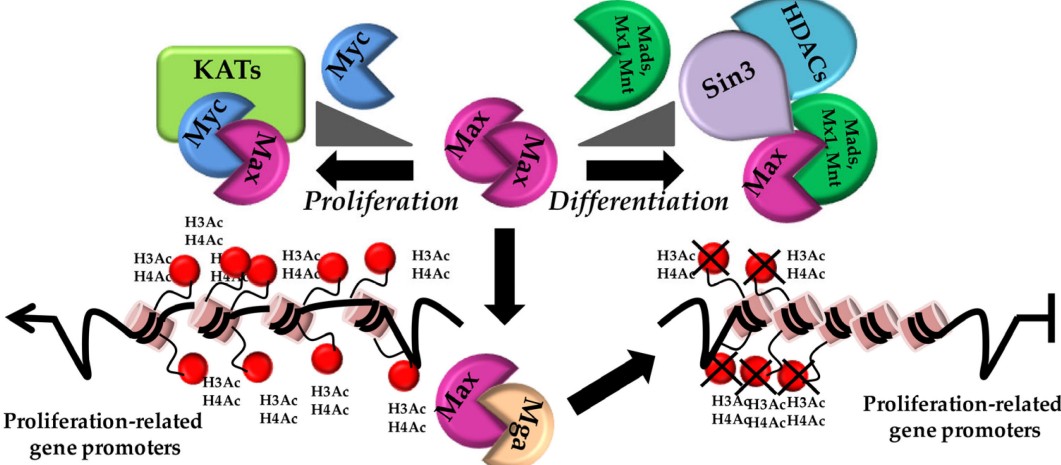

**Figure 2.** The Myc/Max network. Picture depicting Max as the central hub for the Myc/Max network. According to the relative balance between Myc levels and Mad/Mx1/Mnt/Mga within a given cell, different heterodimers are formed, dictating chromatin state (condensed vs. relaxed), repression/transcription of different gene sets and cell fate.

### 2.2. Myc Transcription beyond DNA Binding

In recent years, the extensive knowledge about novel Myc functions led researchers to reconsider its role as a "classical" TF, promoting transcription through DNA binding and histone lysine acetyltransferases (KATs) recruitment, making chromatin permissive to the

loading of the basal transcriptional machinery. Myc regulates all three RNAPs and controls transcripts output of mRNAs, non-coding(nc)RNAs, tRNAs and rRNAs. Importantly, Myc tightly controls RNAPII. In the last decade, the finding that Myc controls transcriptional pause–release was a milestone in the comprehension of some aspects of Myc-dependent transcription. By ChIPseq analyses, Rahl et al. were able to demonstrate that inhibiting Myc/Max heterodimer formation with the 10058-F4 small molecule caused a reduction in RNAPII in the transcribed regions of Myc-dependent gene in ES, with a consequent increase in RNAPII travelling ratio (TR) [18]. Immediately after, by high-throuhgput ChIPseq and RNAseq analyses, a number of works have revealed Myc as an amplifier of transcriptional programs already active or poised in normal and cancer cells [46–48]. This activity also depends on the ability of Myc to invade chromatin, even when the binding affinity is low. The more Myc is expressed, the more binding sites (both high- and low-affinity) are saturated [49]. Open chromatin at active promoters is a pivotal feature in Myc-bound DNA elements [50]. Further, enhancer loops adjacent to gene core promoters may put Myc in close proximity to other enhancers, when promoters are already saturated. More recently, Myc has been found to even associate with and activate topoisomerases 1 and 2 to sustain torsional stress due to high rates of transcription occurring during the amplification process [51]. Myc also controls RNAPII elongation [19], maintaisn the fidelity of splicing [21] and is involved in transcription termination through the Protein Arginine Methyltrasferases 5 (PRMT5) [20]. Myc also regulates mRNA capping by TFIIH kinase recruitment, RNAPII phosphorylation and cap RNA methyltransferase (RNMT) activation [52]. Moreover, Myc controls transcription of indirect target genes by regulating the expression of specific miRNAs, as also revealed by miRnome analysis of Myc-interfered cells [23,53].

## 3. The Role of Myc in Pluripotent Stem Cells

The fundamental importance of Myc for the maintenance of pluripotent cells stemness is proven by the evidence that *c-myc* knock out is embryonic lethal at days 9.5–10.5 of gestation, due to major deficiency in hematopoiesis and vascular ontogeny [54]. Furthermore, although Myc and N-Myc play redundant roles during pre- and early post-implantation stages of embryogenesis [55,56], at least one of them is required for the correct execution of early developmental programs in mouse ES [57,58].

Myc action in pluripotent stem cells is not limited to the control of self-renewal and proliferation potential, but is exerted also at the metabolic level and controls diverse signaling pathways. Overall, all these Myc activities may be attributed to the impact Myc has on the epigenome as a whole. In this section, highlights as to the multiple roles Myc plays in pluripotent stem cells are provided.

### 3.1. Myc and the Stem Cells Epigenome

Pluripotent stem cells' epigenome is highly dynamic and characterized by an open chromatin conformation, which responds to two fundamental requirements: it has to be malleable to activate any developmental transcriptional program and has to possess a memory to recall its pluripotent state [59–61]. Stem cells' chromatin typically shows histone marks of gene activation—i.e., acetylated histone H3 and H4 (H3Ac, H4Ac) and methylated histone H3 on lysine 4 (H3K4me3)—and has the unique feature to simultaneously present, on developmental gene promoters, histone marks of both transcriptional activation and repression, specifically, H3K4me3 and H3K27me3, respectively. This specific signature, the so-called bivalent mark, makes the downstream gene "poised" for activation or repression according to the chromatin reader recruited upon the appropriate stimulus [62,63]. Myc accounts for stem cells' epigenome plasticity in a number of different ways. Its genomic location mirrors chromatin hyperacetylation and H3K4me3-enriched domains; indeed, Myc depletion results in a 70% decrease in H3K4me3 [64–66]. Furthermore, Myc directly controls the expression of chromatin remodelers such as members of the Polycomb repressive 1 and 2 (PRC1/2) complexes and Switch/Sucrose-non fermentable (SWI/SNF) complexes [67,68],

interacts with KATs [66,69] and co-localizes with histone demethylases [70]. At the same time, it also represses the expression of developmental genes, interacting with co-repressors such as Sin3a and 3b, HDAC1/3 and LSD1 [71–73]. More importantly, by interacting with PRC2 and due to the overexpression of Suz12—a component of the PRC2 complex—it allows a massive deposition of H3K27me3 repressive marks on differentiation-related genes [74].

Inducible, long term overexpression of exogenous Myc makes ES independent from leukemia inhibitory factor (LIF). In these cells, wnt signaling induces endogenous *myc* and *nmyc* genes to maintain their pluripotency-activated programs, even after exogenous Myc deprivation, in a positive feedback loop, allowing ES cells to fully rely only on Myc for their stemness [74]. According to the reversible nature of epigenetic mechanisms [75], Myc-/Wnt-dependent ES fully reprogram to a JAK-/STAT3-regulated cascade upon LIF exposure, with a global change in the epigenetic state of chromatin regions regulating the expression of genes related to both pathways. These also evidence the role of Myc in the regulation and maintenance of stem cells' epigenetic memory.

*3.2. Myc in ES*

As anticipated above, *myc* and *n-myc* are required for early development in ES. Simultaneous *myc* and *n-myc* deficiency causes premature ES endoderm and mesoderm differentiation [76]. Indeed, it is now widely accepted that Myc-dependent self-renewal of ES relies on its ability to suppress early differentiation programs. Myc exerts this function both directly and indirectly, regulating both Polycomb complexes and the expression of differentiation-related genes [66,74,77]. *Myc*-deficient cells arrest their development to the early progenitors' stage and do not express markers of terminal differentiation [58]. This function of Myc is also bolstered by the finding that pluripotency markers, such as Oct-4 and Nanog, are Myc-independent. In fact, they continue to be expressed during early stages of development when they co-exist with differentiation markers [78], constituting a transcriptional module—the "core module"—well-defined with respect to the "Myc-module" [69]. The latter has been identified by using the in vivo metabolic biotin tagging method—used to tag well-known Myc and both direct and indirect Myc interactors—followed by mass spectrometry coupled with bioChIP-chip experiments, performed to underscore the genomic targets of the whole module. This Myc-centered cluster of TFs and chromatin remodelers, which includes the Tip60–Ep400 complex [79], does not interact with the core module and is located onto distinct, acetylated histone H3 (H3Ac)- and H4 (H4Ac)-enriched genomic loci. Of note, the Myc module occupies more genomic targets in ES than the core module, indicating Myc as a master regulator of ES transcriptional programs [69]. Specifically, 8000 chromatin elements have been found to be bound by Myc in ES, clearly different from genomic loci loaded with the core module factors [72]. This huge number of Myc-enriched chromatin sites suggest that, in addition to its direct role related to chromatin unwinding ability and RNAPs recruitment, in leading to transcriptional activation, Myc may act as a hot spot for the integration of different gene networks which ultimately grant ES pluripotency. Indeed, genes involved in cell cycle, metabolic and growth regulation belong to the Myc module [69,72]. However, an exception is represented by Myc direct regulation of Sox2 [66] (a component of the core module) and Gata6 (a determinant TF of endoderm differentiation) [57].

Other direct Myc targets in ES cells are cell cycle regulators. ES usually present a fast cell cycle progression and spend 50–60% of their time in S phase. This characteristic is consistent with the permanent expression of cycline-dependent kinases (CDKs) [80,81], with lack of retinoblastoma (RB) [82] and CDKs inhibitors (CDKi) such as p21 and INK4A [80,83]. Cell cycle regulators—including cyclins, CDKs and E2F—are regulated by Myc in ES either at the transcriptional or post-transcriptional level [48,68,72,77,84]. Further, Myc inhibits the expression of CDKi, competing with Miz-1 for chromatin binding, recruiting chromatin-remodeling enzymes or modulating the expression of other regulators [84,85].

In this latter regard, Myc binds to the promoter and upregulates mir17-92 cluster, which target p21, RB2 and cyclin D1 [57,77], forcing ES to proliferate.

All these Myc activities account, at least in part, for the maintenance of ES' typical cell cycle structure, with very short G1 phase, impeding the start of differentiation by limiting chromatin access to chromatin remodelers [86–88].

As for the cell cycle, Myc also controls ES metabolism, which mainly depends on aerobic glycolysis [89], also known as the Warburg effect, a phenomenon extensively investigated in cancer cells [90]. Myc basically regulates any metabolic pathway, from glycolysis and glutaminolysis to nucleotides, 'lipids' and amino acids' synthesis [91,92]. It has been demonstrated that, upon mir290-mediated derepression, Myc directly binds the promoters of *pyruvate kinase M2 (PKM2)* and *lactate dehydrogenase A (LDHA)* genes, allowing the glycolytic metabolic switch in ES [93]. Furthermore, since Myc modulates the level of metabolic intermediates such as $\alpha$-ketogluatarate, acetyl-coenzyme A, nicotinamide adenine dinucleotide, flavin adenine dinucleotide and S-adenosylmethionine [10–12,94], all known to be associated with epigenetic activity [95–98], it is supposed to orchestrate epigenomic changes—represented by both histone acetylation and histone/DNA methylation—in ES, determining their fate across stemness maintenance or differentiation.

### 3.3. Myc and Somatic Cell Reprogramming

In the early 2000s, the work of Yamanaka's lab [16] opened a new research avenue with large potential and enormous impact in the field of regenerative medicine. In fact, the generation of pluripotent stem cells from human fibroblasts theoretically became the basis for the reconstruction of any kind of damaged tissue. From then on, induced pluripotent stem cells (iPS) have been generated from many adult, tissue-specific cells [99,100] and have been induced to follow a wide variety of differentiation routes [101–106]. Substantially, Yamanaka's protocol relies on the ability of defined TFs to revert the heterochromatic somatic cell epigenome into a euchromatic state typical of pluripotency (see Section 3.1). Oct4, Sox2, KLF-4 and Myc (OSKM) are sufficient to produce stem cells from somatic cells through a multistep program in which Myc and OSK modulate each other [107]. During this process, Myc, on the one hand, makes chromatin permissive to OSK binding and, on the other hand, represses fibroblast-specific genes and binds to pluripotency genes enhancers, strengthening reprogramming [108,109]. However, Myc requires OSK chromatin binding to exert its function in a positive feedback loop [109]. Early reports on the many attempts to substitute Myc in the Yamanaka's cocktail, due to its oncogenic potential [110], displayed that reprogramming results were not very efficient [111,112]. However, a few years later, the use of L-Myc and of Myc mutants lacking transforming activity, but retaining Max binding properties, was found to be even more efficient than wild type Myc in generating iPSC colonies [113]. Very recently, the precise role of Myc during somatic cell reprogramming has been defined. Homogenous populations of mouse fibroblasts were depleted of methyl-CpG binding domain protein 3 (Mbd3) and GATA zinc finger domain-containing 2a (Gata2da), both participating in the Nucleosome remodeling and deacetylase (NuRD) repressor complex [114]. In this cell system, whereas OSK factors dynamically bind overlapping enhancers throughout the process, Myc preferentially binds to promoters. Specifically, Myc binds constitutively active promoters' genes (CAPGs), igniting biosynthetic pathways, such as DNA synthesis and repair, cell proliferation and chromatin dynamics. In this system, OSKs are sufficient for somatic reprogramming to occur until endogenous Myc is expressed. Upon depletion of all three paralogs, OSKs fail to reprogram mouse fibroblasts. However, the opposite—i.e., OSK deprivation and Myc overexpression—promotes the expression of genes driven by CAPGs and the repression of somatic genes', but not pluripotency genes', induction. Of note, although Myc has been invariably associated with histone modifications related to gene activation (see above), CAPGs do not present the chromatin-switching typical of pluripotency genes upon reprogramming, but, as in cancer cells [115], are regulated by changes in the tRNA pool and their preferential codon usage. During reprogramming, a shift in the codon usage by tRNA is observed. In particular,

CAPG-activated genes present a shift from G/C ending codon to A/T, whereas repressed genes shift A/T ending codon to G/C. This phenomenon corresponds to the enrichment of activating histone marks (i.e., H3K4me3) in chromatin regions next to tRNA encoding genes, fulfilling the request of enhanced elongation and translation of transcripts during reprogramming [116]. Given Myc-dependent regulation of RNAPIII activity [117,118], it may be hypothesized that Myc controls somatic cell reprogramming by directly binding to CAPGs and amplifying elongation and translation of the corresponding gene products.

## 4. Mammalian Tissue Regeneration: A Transcriptional Point of View

Transcriptional profiles activated during tissue regeneration closely resemble embryogenesis-related programs which drive proliferation, migration and cell morphogenesis in different organisms, including humans [119–121]. Of course, transcriptional signatures in regenerating cells are the result of the intersection of multiple, orchestrated layers of control; indeed, DNA elements, transcription factors and epigenetic machineries altogether participate in the constitution of networks that rule the execution of programs which may diverge from one tissue to another. In some organisms, as stated above, embryonic-like profiles are found [122,123], but this is not a dogma, as, in other species, transcriptional signatures mostly mirror those activated during injury and/or damage [124,125]. Furthermore, as expected, these regeneration-specific signatures are affected by environmental cues, mainly inflammation and immunometabolism [126].

It has to be noted that higher vertebrates, with the exception of African spiny mice [127,128], possess limited regenerating capacity. This has led to the hypothesis that, during evolution, some regeneration-specific chromatin regulatory elements have been lost. In particular, tissue regeneration enhancer elements (TREEs) [129], which control regeneration in zebrafish, salamanders and *Drosophila melanogaster* [130–132], have not been identified yet in mammalians. Indeed, strategies to transfer zebrafish TREEs, combined with CRISPR/Cas-dependent genome editing, to enhance regeneration capacity upon tissue injury have been recently described in small and large mammalians [133]. However, intronic elements have been found to potentially substitute TREEs in mammalians. In fact they control the expression of GATA2 [134] and SAMD14 [135] during hematopoietic cells' regeneration following anemia in mice.

Trans-acting elements, that is, TFs devoted to the maintenance of pluripotency and/or acting during mammalian development, also act as master regulators of regeneration. In particular, SOX2 and KLF4, two of the OSKM factors, have been involved in the regeneration of olfactory neurons and retinal ganglion in mice [136,137]. ETS, SOX and GATA factors have been found to regulate neo-osteogenesis upon jaw distraction in mice, whereas developmental genes such as RUNX2 and DLX5 are repressed [121]. The same occurs in the mouse heart: a number of tissue-specific TFs, including Meis1 [138], Yes-associated protein (YAP) [139] and the Thyroid Hormone Receptor $\alpha$ (THRA) [140], constrain heart regeneration.

Strictly related to TFs' action, chromatin accessibility represents a major characteristic of genomic regions involved in regeneration. This is also in line with a phenomenon known as "hypertranscription", characterized by an increase in the nascent transcript output of the majority of the transcriptome, in particular, of housekeeping and ribosomal genes, which specifically respond to the biosynthetic needs of highly replicating cells [141,142]. Although not investigated in detail in mammalian regenerating tissues, it is highly probable that this phenomenon occurs during vertebrate regeneration, as an increase in total RNA content has been found in hematopoietic [143] and small intestine progenitors [144] during physiological regeneration. However, chromatin marks of regeneration are different from those appearing during embryogenesis, regardless of the similar final biological outcome. For example, during zebrafish cardiogenesis, the KAT p300 deposit H3K27Ac marks throughout cardio-specific genes [145], but the same phenomenon does not occur during regeneration [146]. Notably, mammalian chromatin de-compaction due to KATs' activity has been mostly reported in pluripotency- and stemness-related genomic loci that are not

directly involved in regeneration processes [147–149]. Nevertheless, chromatin acetylation has been found to be involved in liver, muscle and neuron regeneration [150–152], whereas histone deacetylases and their inhibition (HDACs) have been implicated in bone and dental pulp mineralization [153,154], axon remyelination [155,156] and neuronal regeneration [157,158].

A central role in re-shaping chromatin topology during regeneration seems to be played by the Polycomb group (PcG) and SWI/SNF remodelers, which may be considered as two sides of the same coin.

Erasure of PcG-dependent H3K27me3 histone marks from enhancers devoted to regeneration has been found both in *Drosophila melanogaster* and zebrafish [132,159], and also in mouse Schwann cells' proliferation and axon regeneration upon the occurrence of damage [160]. A direct role of PcG complex members has been found in neurons [161], bladder [162], heart [163] and skeletal muscle [164].

SWI/SNF activity has been found to be essential for liver regeneration [165]. However, in certain conditions, SWI/SNF negatively controls the regeneration process. Indeed, the SWI/SNF component AT-rich interactive domain-containing protein 1A (ARID1A) has been found to impair Hippo-dependent regeneration [166,167] by sequestering YAP and TAZ from their target, TEAD1, the lead factor regulating regeneration downstream from the Hippo pathway [167]. Under mechanical stress, which may occur during regeneration [168], nuclear F-actin prevents the binding of AIRD1A to YAP/TAZ, allowing the latter to activate TEAD1 and Hippo-dependent transcription [169].

## 5. Myc Action in Tissue Regeneration

That Myc may exert pivotal functions in the regeneration of tissues is almost obvious. Indeed, both quiescent stem cells' activation and dedifferentiation of adult, tissue-specific cells foresee the ignition of silent transcriptional programs which are largely dependent on Myc protein activity. As occurs during somatic cell re-programming [116], Myc modules have also been identified in physiologically regenerating cells and upon tissue injury. In contrast to what is observed during carcinogenesis [69], Myc action is tightly regulated in this context and turned off once no longer necessary. In this paragraph, examples of Myc function in the regeneration of different tissues are provided.

### 5.1. Myc in Liver Regeneration

Myc is rapidly induced upon partial hepatectomy (PH), as it is necessary for the G0/G1-S transition of the cell cycle in hepatocytes. Elegant experiments performed in transgenic mice where *myc* was conditionally knocked-down have also shown that Myc controls hepatocyte ploidy and size, being involved in multiple sequential cell cycles not followed by cytokinesis, and, although dispensable for adult hepatocytes proliferation, it is essential for liver regeneration after PH [170]. Knock-out of *myc* and of its partner *mlx* completely blunts liver regenerative capacity, affecting signaling pathways involved in mRNA translation, hepatic steatosis and mitochondrial function; KO livers show characteristics typical of non-fatty alcoholic liver disease (NFALD) [171]. At the molecular level, it has been described that PH-induced Inhibitor of DNA binding (or inhibition of differentiation) 2 (Id2)—another well-known Myc functional partner [172–174]—detaches and sequesters transcriptional co-repressors from *myc* promoter, leaving E2F4 and p130 pocket protein on, activating Myc transcription. In turn, after a few hours, Myc activates Id2 transcription, allowing the repositioning of this protein together with co-repressors onto *myc* promoter, in a tightly regulated feedback loop [175]. Furthermore, a Jun/Myc/BLC2/CCND1 axis has been found to regulate liver regeneration upon PH due to the downregulation of mir429 [176].

It has to be mentioned that Myc may exert a detrimental role in the liver, further to its well-established tumorigenic effect. Indeed, it may induce liver fibrosis upon PH in *gata4*-deficient hepatocytes by inducing the expression of the platelet-derived growth

factor-β (PDGFRβ), transdifferentiation of endothelial cells from sinusoidal to continuous and activation of hepatocyte stellate cells [177].

### 5.2. Myc Role in Pancreatic β-Cells' Replacement

In pancreatic β-cells, Myc is regulated by glucose and is probably part of the proliferation adaptive machinery in these cells [178]. Further, it is required for neonatal β-cells' specification [179], leading to the exciting hypothesis of using Myc overexpression to counteract β-cells' loss. However, years ago, the possibility of recovering pancreatic β-cells' dysfunction by Myc overexpression was discarded, as supraphysiological levels of Myc induced cell death and differentiation [180–182]. Nonetheless, although Myc is dispensable for adult β-cells' proper function in healthy conditions, in situations of metabolic stress, Myc has been revealed to be necessary to counteract the impairment in glucose tolerance, hyperglycemia and ipoinsulinemia [179,183]. Therefore, whereas high levels of Myc are detrimental, a fine-tuning of its expression may be beneficial to dysfunctional β-cells [184].

### 5.3. Myc in the Regulation of Intestinal Crypts' Stemness and Regeneration

Wnt-dependent Myc expression has been found to be essential for the proliferation of progenitor cells in the intestine and is expressed in the amplifying compartment of intestinal crypts during development; however, it appeared to be dispensable for juvenile and adult intestine homeostasis [185,186]. Myc-depleted crypt progenitors do not exit the cell cycle but progress more slowly and are smaller in size, indicating the fundamental biosynthetic role of Myc in progenitor cells [187]. In addition to this developmental role, Wnt-/Myc-signaling plays a role during intestine regeneration upon DNA damage via upregulation of Focal Adhesion Kinase (FAK). Indeed, *apc* deletion produces an upregulation of FAK, which reverts to its normal level upon *apc/myc* double depletion. Moreover, upon *myc* conditional knock-out, intestinal crypts' regeneration after gamma irradiation is severely impaired, with few enlarged regenerating crypts and a huge number of cystic areas. Highly similar results are obtained upon *fak* deletion [188]. In agreement with these findings, a role of the PCNA Associated Factor (PAF)-mediated Myc activity in intestinal progenitor cells upon injury has been found. Specifically, PAF has been found located onto the *myc* promoter, leading to its transactivation and intestinal regeneration due to progenitor cells' self-renewal. Remarkably, PAF knock-out impairs both intestinal regeneration and Apc depletion-induced tumorigenesis, along with Myc downregulation [189].

### 5.4. The Role of Myc in Cardiomyocytes' Proliferation

Terminally differentiated tissues—such as the heart and the central nervous system (CNS)—are refractory to Myc-dependent proliferation signals. Indeed, it has been demonstrated that, whereas in permissive tissues, such as liver, Myc-bound promoters can be activated by mitogenic stimuli, in non-permissive tissues, Myc activity is locked even in the presence of such signals, probably because of limited availability of components of the basal transcriptional machinery in which Myc is deployed to exert its function [190]. Supporting this hypothesis, overexpression of cyclin T1, which, together with cyclin-dependent kinase 9 (CDK9), constitutes the P-TEFb complex [18], in cardiomyocytes, unlocks the proliferative capacity of Myc in vivo, inducing the transcription of a subset of Myc targets normally expressed in the liver but not in the heart and cardiomyocytes (CMs). More importantly, when Myc was transiently induced in a cyclin T1 overexpressing context, no consistent and detrimental effects on long-term survival were detected [191]. In the heart, Myc is also engaged in a positive feedback loop with the lncRNA small nucleolar RNA host gene 1 Snhg1 (Snhg1), promoting cardiomyocytes' proliferation and regeneration upon both myocardial infarction and ischemia/reperfusion, with reduced CMs' apoptosis and size of the infarcted area. Mechanistically, Snhg1 induces phosphatase and tensin homolog (PTEN) degradation and phosphatydilinositol 3-kinase (PI3K)/protein kinase B (AKT) activation which, in turn, via Glycogen synthase kinase-3β (GSK3β), promote Myc stabilization. Myc occupies and activates *Snhg1* promoter in a self-sustaining loop, leading to CMs' proliferation [192].

Moreover, the ability of Myc to grant reprogramming of adult somatic cells has also been successfully tested in cardiomyocytes, with different outcomes according to the timing of its expression. Indeed, whereas doxycycline-inducible, short-term expression of OSKM opens a transient transcriptional window of dedifferentiation with the expression of α-smooth muscle actin (α-SMA) and slightly reduces left ventricular ejection fraction (LVEF) and stroke volume (SV), prolonged OSKM exposure leads to an extensive heart remodeling, to a deterioration of cardiac function in the middle period and to the onset of cardiac tumors in the long period, even after doxycycline withdrawal. These results indicate the full epigenetic reprogramming of CMs in vivo [193].

### 5.5. Myc and the Regeneration of the Nervous System

The nervous system (NS) has evolved as a protected closed district—unresponsive to proliferative signals [194]—to preserve synapses and strengthen memory and character. These features are dependent, at least in part, on changes in the NS transcriptome, which is epigenetically preserved to express neuron-specific transcripts and is refractory even to OSKM reprogramming [195]. Accordingly, Myc overexpression in the NS provides opposite results when compared to other tissues, where it leads to carcinogenesis. Indeed, its expression is blunted in a number of ways in the NS and it, surprisingly, induces apoptosis and neuronal loss preceding and during neurodegeneration. Phosphorylated Myc has been detected in brain tissues affected by tauthopathies, including supranuclear palsy, Alzheimer's Disease (AD) and corticobasal degeneration [196,197] and has been linked to both Ras/MAPK [196] and PI3K/mTOR pathways [198,199]. Derangement of the cell cycle is typical of both cancer and neurodegeneration, but, whereas in the first condition, hyperproliferation is the major outcome, neurons abort the cell cycle before DNA synthesis [200] or, rarely, after G2/M phase [194]. These two opposite phenomena, despite a common molecular background, seem to rely on the entity of the stress to which neuronal cells are exposed. Upon acute and high-energy stress exposure, neurons activate mTOR and ROS production, with the consequent engagement of the Forkhead box (FOXO) family of transcription factors, leading to the production of anti-oxidant enzymes, protecting neurons from death [201,202]. FOXO proteins compete with Myc for binding to cell cycle gene promoters [203,204]. Upon chronic stress stimuli, the FOXO protective role is bypassed and Myc activities are not repressed [205]. Furthermore, Myc is downstream from an NMDA receptor-dependent apoptotic cascade, which also comprises NF-kB and p53 activation as the final event [206].

Thus, it may seem that Myc exerts a potent detrimental role in the NS. This is not true, as Myc plays an important function during neuronal cells' repair upon injury. However, there is a clear dichotomy regarding Myc reparative action between the peripheral and the central NS (PNS and CNS). In the PNS, Myc is re-activated upon axotomy in adult dorsal root ganglion [207] and supports the regeneration of sensory axons. In this latter regard, mechanistically, Myc activates TERT subunit of telomerase which, in turn, activates p53. This Myc/TERT/p53 pathway appears to play a major role in axon regeneration, acting both on metabolism and mitochondria, regulating ROS production [208]. Notably, when axotomy is performed on CNS elements, such as the optic nerve, Myc levels decrease even in uninjured neurons which undergo cell death [207]. The nature of the mechanism controlling the differential expression of Myc in PNS and CNS has to be elucidated, but probably depends on the strict relationship between the levels of FOXO protein and Myc [209,210].

### 6. Conclusions and Translational Perspectives

Far from being exhaustive, this review article provides highlights on the role of Myc in both the physiological regeneration and recovery upon damage in main mammalian tissues, such as liver, intestine, pancreas, heart and the CNS. As expected, other tissue districts deploy Myc expression as a tool to regenerate, repair and counteract cellular senescence. Indeed, Myc has been found to promote basal epithelial cells' self-renewal in the esophagus,

counteracting ROS-dependent aging processes [211], and to participate in the recovery of impaired diabetic wound healing downstream from the silent information regulator 1 (SIRT1) signal [212]. When looking at the possibility to employ Myc as a rejuvenating factor, somatic cell reprogramming represents the most logical strategy to obtain tissue regeneration even in high-proliferation signals' refractory tissues. Partial OSKM-dependent reprogramming of myofibers, for example, has been employed to modify satellite stem cells' niche epigenetic state through downregulation of Wnt4 signaling and to induce muscle regeneration [213], whereas co-activation of exogenous Myc and Notch results in the reprogramming of different inner ear cell types, including hair cells, whose loss is responsible for hearing loss in humans. It has to be noted, however, that, in this latter case, only a transient expression of Myc/Notch promotes hair cells' transdifferentiation from supporting cochlear cells [214].

More importantly, it has been recently reported that cycles of transient expression of the OSKM factors slows the appearance of senescence markers and increases life-span in a mouse model of premature aging by modifying the cell epigenome. Moreover, OSKM factors improve cell recovery upon metabolic disease and muscle injury in old mice, indicating a comprehensive amelioration of the age-associated phenotype based on epigenetic mechanisms [215].

Although exciting, this perspective is far from being the elixir of life. In fact, the expression of OSKM factors—especially Myc—has to be tightly controlled in space and time to avoid highly dangerous side effects, such as the development of tumors (Figure 3).

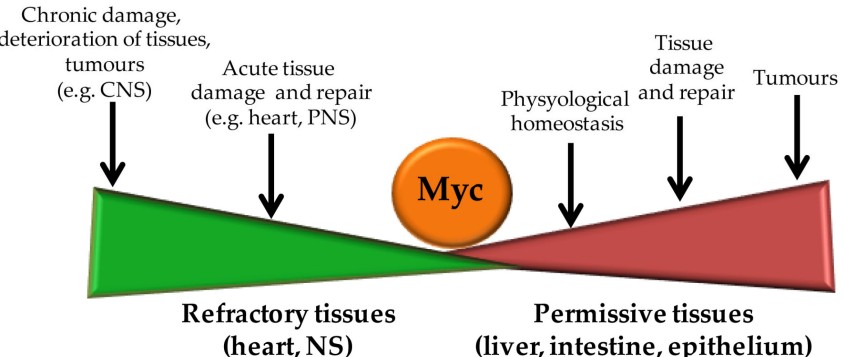

**Figure 3.** Biological outcomes of increased Myc protein levels. Myc levels are tightly controlled within the cell. It may increase in narrow windows of expression upon mitogenic stimuli ruling the physiological homeostasis of highly regenerating permissive tissues and upon tissue damage. In highly terminally differentiated tissues, refractory to proliferation signals, Myc levels increased upon both acute and chronic tissue damage, with different outcomes. Indeed, either repair or tissue degeneration may occur. As well established, sustained and uncontrolled Myc levels promote its oncogenic properties.

Associated with this consideration, it has to be underlined that the field of cell-based regenerative medicine presents a number of challenges that have to be solved. First, the choice of vectors to be used has to be carefully weighed [216]. Currently, viral non-integrating episomal vectors and transposons are used to safely generate iPS [216,217]. Furthermore, each step to translate a cell-based therapy proof-of-concept into the clinic has to be tightly controlled. First of all, standard criteria and associated information should come with the release of expanded cells (e.g., stage of differentiation, phenotype, delivery system, factors to provide and granting the maintenance of the phenotype). Linked to this issue is manufacturing. Bulk, automated manufacturing arising from small pilot labs is highly expensive and requires bioreactors which, however, get into the game only when evidence of clinical efficacy has been provided and when extensive funds are available. Further, the pace of science in the field frequently makes the current manufacturing protocols and technologies obsolete. This requires continuous updates and advancements to satisfy the latest innova-

tions, such as the direct expansion of cells on 3D-porous scaffolds or their assembly through bioprinting. Biomaterials, including starting fetal bovine serum-deprived cell culture media, represent another important issue. Extracellular matrix-derived substances, biopolymers and hydrogels are the most relevant, likely providing environmental responsiveness [218]. Financial profit is a major problem, as it underlies the commercialization of poorly characterized cell products with uncertain benefits by companies which leverage patients with no expectancy of cure. This highlights the importance of robust regulatory procedures and guidelines, ensuring evidence-based production of cell-based products for therapeutic purposes [219]. In spite of this evidence, somatic cell reprogramming experiments have opened a new avenue in the search of platforms—constituted by TFs and small-molecule modulators of chromatin remodeling enzymes, for example—which are suitable to epigenetically remodel adult and old genomes to fight age-associated and degenerative diseases.

**Author Contributions:** B.I. and S.N. wrote and revised the entire manuscript. All authors have read and agreed to the published version of the manuscript.

**Funding:** This work has been partially funded by the Italian Association for Cancer Research grant IG15927 to B.I.

**Conflicts of Interest:** The authors declare no conflict of interest.

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
