# Peer review of "Myc beyond Cancer: Regulation of Mammalian Tissue Regeneration"

_pathophysiology, doi:10.3390/pathophysiology30030027_

Round 1

Reviewer 1 Report

1.The review article "Myc beyond cancer: regulation of mammalian tissue regeneration" explores the role of Myc in tissue regeneration, highlighting its implications not only in cancer research but also in aging and regenerative medicine. The article emphasizes the multifaceted functions of Myc beyond its involvement in tumorigenesis.

The c-myc gene is responsible for encoding the Myc protein, which possesses various domains that govern transcriptional activation, repression, protein interactions (such as with Max), and DNA binding to specific sequences. Understanding the structural features of Myc provides insights into its diverse functions in regulating gene expression and cellular processes.

The review also focuses on Myc's roles in pluripotent stem cells, including its involvement in maintaining self-renewal and proliferation capabilities, regulating metabolic processes, and controlling signaling pathways. These functions of Myc are interconnected and contribute to the maintenance of pluripotency in stem cells.

Furthermore, the article discusses how the transcriptional profiles and chromatin dynamics observed during tissue regeneration exhibit similarities to those seen during embryonic development. However, these profiles can vary depending on the specific regenerating tissue and the environmental cues present. Transcription factors, chromatin remodelers, and epigenetic modifications play pivotal roles in orchestrating the regeneration process. Gaining a deeper understanding of these mechanisms provides valuable insights into the regenerative potential and limitations in different organisms.

Overall, the review article underscores the importance of studying Myc's involvement in tissue regeneration, as it has implications for the development of regenerative therapies and advancements in the field of regenerative medicine. Myc's multifaceted functions and its impact on transcriptional regulation and cellular processes make it a promising avenue for exploration in the context of tissue regeneration.

2. A constructive conclusion should be added at the end of the article.

3. Latest techniques used for the studies should be incorporated. 

The English language is good.

Author Response

MS 2516277 Authors’ responses to Reviewer 1

The authors wish to thank the Reviewer for the positive evaluation of their MS and constructive remarks. Below are our point-to point answers.

Q1. A constructive conclusion should be added at the end of the article.

A1. We thank the reviewer for this comment. A conclusion section was already present as section 6 in the previous version of the MS. In the revised MS, we have added a paragraph related to the challenges the field of regenerative medicine presents. We believe that the new Conclusions section is now more constructive and improved. Changes to this specific section are tracked in red colour.

Q2. Latest techniques used for the studies should be incorporated

A2. The more important techniques used to perform the studies cited in this review article are now cited and shown in blue colour in sub-paragraphs 2.2 and  3.2. Basically, the genome wide studies reported in our work were approached with well established hightroughput techniques, such as ChIPseq, RNAseq, ChIP-chip, mass spectrometry.

Reviewer 2 Report

The review titled “Myc beyond cancer: regulation of mammalian tissue regeneration” describes in detail the role played by Myc in stem cells pluripotency. The authors have discussed the pathways involved and factors affecting the Myc-mediated activity, and examples of Myc-mediated tissue regeneration. This review will be very useful in understanding one of the non-cancer related, essential role that Myc plays. See below a few comments to enhance reading experience of the manuscript.

·       It would be useful to add the following reference that describes use of transformation-deficient Myc. https://www.pnas.org/doi/abs/10.1073/pnas.1009374107.

·       In the Conclusions and translational perspectives section, it would be useful to comment on the challenges due to disordered Myc structure and on how researchers are addressing the issue.

·       What are some of the challenges of regeneration medicine in general, in addition to the expression of OSKM factors that needs to be tightly controlled?

·       Line 75: “positive”

·       Check throughout the document and keep space before reference annotations consistent. Eg. Lines 140 vs 144.

·       Line 161, ES already defined in the document in the introduction.

·       Line 192 and 196: “Wnt”, ‘W’ should be capitalized.

·       Line 195: “stemness”

·       Line 207: “Myc”, ‘M’ should be capitalized since it is beginning of a sentence.

·       Line 255: Sentence needs reconstruction. For example: “In the early 2000s’ the work of Yamanaka’s lab [16] opened a new research avenue with large potential and enormous impact..”

·       Line 257: “theoretically became the basis for..”

·       Line 264: “modulate each other.”

·       Line 266: “reprogramming results were not very efficient”.

·       Rephrase line 274.

·       Line 293: “maybe hypothesized that..”

·       Line 313: “Drosophila Melanogaster”, italicized.

Needs moderate editing to fix grammatical errors.

Author Response

MS 2516277 Authors’ responses to Reviewer 2

The authors wish to thank the Reviewer for the positive evaluation of their MS and helpful criticisms.

Below are our point-to-point answers. Changes are tracked in red colour.

      Q1. It would be useful to add the following reference that describes use of transformation-deficient Myc. https://www.pnas.org/doi/abs/10.1073/pnas.1009374107.

A1. We thank the Reviewer for this suggestion. The reference (113) has been added in sub-paragraph 3.3.

Q2. In the Conclusions and translational perspectives section, it would be useful to comment on the challenges due to disordered Myc structure and on how researchers are addressing the issue.

A2. We thank the Reviewer  for having raised this important issue. We hope the Reviewer does not mind if we have discussed this important topic in section 2. In our opinion, it better fits in this paragraph.

Q3. What are some of the challenges of regeneration medicine in general, in addition to the expression of OSKM factors that needs to be tightly controlled?

A3. We thank the Reviewer for this comments. We hope we catched the point. We have added a paragraph on the challenges the field of regeneration medicine presents in general in the Conclusion section.

Q4.Line 75: “positive”

A4. We thank the reviewer. We have corrected.

Q5. Check throughout the document and keep space before reference annotations consistent. Eg. Lines 140 vs 144.

A5. We have checked spaces before references.

Q6. Line 161, ES already defined in the document in the introduction.

A6. Corrected

Q7.  Line 192 and 196: “Wnt”, ‘W’ should be capitalized.

A7. Corrected

Q8.  Line 195: “stemness”

A8. Corrected

Q9. Line 207: “Myc”, ‘M’ should be capitalized since it is beginning of a sentence.

A9. Corrected

Q10.  Line 255: Sentence needs reconstruction. For example: “In the early 2000s’ the work of Yamanaka’s lab [16] opened a new research avenue with large potential and enormous impact..”

A10. The sentence has been reconstructed

Q11. Line 257: “theoretically became the basis for..”

A11. Corrected

Q12. Line 264: “modulate each other.”

A12. Corrected

Q13. Line 266: “reprogramming results were not very efficient”

A13. Corrected

Q14. Rephrase line 274

A14. Line 274 has been rephrased

Q15. Line 293: “maybe hypothesized that..”

A15. Corrected

Q16. Line 313: “Drosophila Melanogaster”, italicized.

A16. Drosophila Melanogaster has been italicized